# Effects of Resveratrol on Mouse B16 Melanoma Cell Proliferation through the SHCBP1-ERK1/2 Signaling Pathway

**DOI:** 10.3390/molecules28227614

**Published:** 2023-11-15

**Authors:** Xiaoke Yu, Zhiyang Sun, Saiya Nie, Tao Zhang, Hongzhao Lu

**Affiliations:** 1School of Biological Science and Engineering, Shaanxi University of Technology, Hanzhong 723001, China; yxk02200059@163.com (X.Y.); 17809268258@163.com (Z.S.); 17342986160@163.com (S.N.); zhangtao780823@snut.edu.cn (T.Z.); 2Qinba State Key Laboratory of Biological Resources and Ecological Environment, Shaanxi University of Technology, Hanzhong 723001, China; 3Department of Biology, QinLing-Bashan Mountains Bioresources Comprehensive Development C. I. C., Shaanxi University of Technology, Hanzhong 723001, China; 4Shaanxi Province Key Laboratory of Bio-Resources, Shaanxi University of Technology, Hanzhong 723001, China

**Keywords:** resveratrol, melanoma, SHCBP1, MAPK/ERK signaling pathway

## Abstract

Melanoma originates from the malignant mutational transformation of melanocytes in the basal layer of the epidermal layer of the skin. It can easily spread and metastasize in the early stage, resulting in a poor prognosis. Therefore, it is particularly important to find effective antitumor adjuvant drugs to inhibit the occurrence and development of melanoma. In this study, we found that resveratrol, a polyphenolic compound from grape plants, can significantly inhibit the proliferation, colony formation and migration of mouse melanoma B16 cells. Notably, resveratrol was also found to inhibit the expression of SHCBP1 in B16 cells. Transcriptional analysis and cellular studies showed that SHCBP1 can activate the MAPK/ERK signaling pathway to regulate cyclin expression and promote the G1/S phase transition of the cell cycle by upregulating ERK1/2 phosphorylation levels. Resveratrol further downregulates the phosphorylation level of ERK1/2 by inhibiting SHCBP1 expression, thus inhibiting tumor cell proliferation. In conclusion, resveratrol inhibits the proliferation of B16 cells by regulating the ERK1/2 signaling pathway through SHCBP1. As an upstream protein of the ERK1/2 signaling pathway, SHCBP1 may be involved in the process of resveratrol-mediated inhibition of tumor cell proliferation.

## 1. Introduction

Melanoma is a common malignant tumor of the skin, with poor prognosis and steadily increasing morbidity and mortality worldwide [1]. At present, the commonly used treatment method for melanoma is surgical resection combined with chemotherapy, but most chemotherapy drugs have limited efficacy due to drug resistance [1,2]. Recently, targeted drug therapies have significantly affected the treatment of melanoma. However, some patients with melanoma are insensitive to targeted therapy drugs, further causing preexisting resistance [2]. Compared with chemotherapy, immunotherapy has low side effects, but the treatment cycle is long and the toxicity of immune drugs is high, which often cannot achieve the purpose of treating cancer as soon as possible [3]. In the past three decades, natural products and their derivatives have accounted for 34% of the drugs approved by the Food and Drug Administration of the United States [4]. There are many targets and pathways of natural products, and the antitumor effect is remarkable, but the mechanism of action is not clear, which often limits the further development and utilization of natural products [4]. Therefore, we chose resveratrol, a polyphenol compound with chemical protective, antioxidant, anti-inflammatory and anti-proliferative activities [5,6], as the research object. Resveratrol as a natural polyphenol is widely used in food, beauty, medicine, and healthcare products [7,8]. Previous studies have demonstrated that resveratrol not only regulates the apoptosis of endovascular cells, photoreceptor cells, Müller cells, and ganglion cells through the NF-κB signaling pathway but also influences the autophagy of cells through TLR4-FOXO3 [9], thus reducing the release of inflammatory factors and the apoptosis of cells [10]. Resveratrol also upregulates the expression and activity of anti-oxidant enzymes to inhibit reactive oxygen species (ROS) production, and consequently attenuates the oxidative stress in cells [11]. In addition, resveratrol inhibits the growth and migration of various malignant tumors by regulating the cell cycle progression and apoptosis [12,13]. However, as a potential adjuvant anti-neoplastic compound, it is unclear whether resveratrol inhibits the occurrence and development of melanoma.

SHC encodes three subunits, including p46Shc, p52Shc, and p66Shc. Each subunit has a carboxy-terminal Src homology domain (SH2), and SH2 domain-containing proteins can dock with phosphorylated tyrosine residues on other proteins. SH2 domains are commonly found in aptamer proteins that contribute to signal transduction in the receptor tyrosine kinase pathway [14]. SHC SH2 domain-binding protein 1 (SHCBP1) is an important connexin on the SH2 domain of the SHC protein, which is highly expressed in many malignant tumors; studies have shown that SHCBP1 promotes the development of cancer through multiple signaling pathways such as FGF, NF-κB, MAPK/ERK, PI3K/AKT and TGF-β1/Smad [15,16,17]. SHCBP1 expression is abnormally elevated in synovial sarcoma (SS), which promotes the metastasis of SS and regulates cell proliferation through the TGF-β1/Smad signaling pathway [18]. Recent studies have demonstrated that the interaction between SHCBP1 and FGF13 could activate the AKT-GSK3α/β signaling pathway and consequently promote the proliferation of the non-small cell lung cancer (NSCLC) cell line A549 [19]. However, whether SHCBP1 is involved in the regulation of the occurrence and development of melanoma remains unknown. In this study, we found that SHCBP1, as an important protein upstream of the ERK1/2 signaling pathway, may participate in the process of resveratrol-mediated inhibition of the proliferation of mouse melanoma B16 cells. This was the first study to reveal the mechanisms underlying the role of SHCBP1 in the regulation of melanoma cell proliferation by resveratrol, and provided theoretical evidence for further investigations into the potential pharmacological mechanisms of resveratrol in influencing the development and progression of other malignant tumors.

## 2. Results

### 2.1. Resveratrol Inhibits G1 to S Phase Transition in B16 Cells

Previous studies have demonstrated that resveratrol has inhibitory effects on the proliferation of various tumor cells [12,13]. In this study, various concentrations of resveratrol, i.e., 20 μM, 40 μM, 60 μM, 80 μM, and 100 μM, were used to assess the influence of resveratrol on the proliferation of B16 cells. The findings showed that with increasing resveratrol concentration, the viability of B16 cells decreased significantly, indicating that resveratrol could inhibit the viability of B16 cells in a dose-dependent manner (Appendix A). Based on the cell growth status, 60 μM resveratrol was selected for the subsequent experiments in this study. Flow cytometry showed that compared with the control group, the number of cells in the G1 phase increased by 1.12-fold, the number of cells in the S phase was reduced by 4%, and the number of cells in the G2 phase was reduced by 8% (Figure 1A and Appendix A). After the formation of cell clones, crystal violet staining showed that clone formation was reduced by 65% in the resveratrol group, compared with the control group (Figure 1B and Appendix A). Western blotting was used to assess the cell cycle-related proteins after resveratrol treatment and showed that the expression of cyclin-dependent kinase inhibitors p21 and p27 increased by 3.2-fold and 1.65-fold, respectively; expression of the tumor suppressor gene p53 increased by 2.2-fold and cell cycle protein Cyclin D1 expression reduced by 85% (Figure 1C and Appendix A). These findings demonstrate that resveratrol inhibited the progression of B16 cells from the G1 phase to the S phase, and blocked the cells in the G1 phase. These results indicate that resveratrol can inhibit the proliferation of B16 cells. In addition, we also found that SHCBP1 expression increased gradually during the growth of B16 cells from 12 h to 72 h (Figure 1D and Appendix A). Interestingly, the expression of SHCBP1 protein was significantly decreased after the addition of resveratrol, indicating that resveratrol could inhibit the expression of SHCBP1 in B16 cells. With an increase in the resveratrol treatment time, the expression of SHCBP1 continued to decrease after 36 h, and the expression of SHCBP1 was the lowest at 60 h. Therefore, we chose 60 h as the treatment time of resveratrol in subsequent experiments. SHCBP1 is involved in intracellular signal transduction and cell division, which is highly expressed in lung cancer, breast cancer, liver cancer and other tumor cells, and promotes tumor cell proliferation. However, the function of SHCBP1 in melanoma cells has not been fully elucidated.

### 2.2. High SHCBP1 Expression Was Beneficial for B16 Cells Proliferation

To further study the effect of SHCBP1 on the occurrence and development of melanoma, we first analyzed the expression of SHCBP1 in normal cells and melanoma cells. The expression of SHCBP1 in mouse melanoma B16 cells was significantly higher than that in MMPC cells (Appendix A). Lentivirus was used to construct the SHCBP1 knockdown B16 cell line, and the findings showed that compared with the NC group, the efficiency of SHCBP1 knockdown was 94% (Figure 2A,B). Meanwhile, the proliferation rate of B16 cells decreased significantly at 12 h, 24 h, 36 h, and 48 h after knockdown of the SHCBP1 expression (Figure 2C). A clone formation assay further confirmed that SHCBP1 knockdown could inhibit the clone formation of B16 cells (Figure 2D and Appendix A). EdU is a thymidine analogue, which can replace deoxythymine and be inserted into DNA molecules during cell proliferation, thus allowing the effective measurement of percentage of cells in the S phase. EdU staining demonstrated that the percentage of positive cells in the S phase was 16% lower in the SHCBP1 knockdown group compared to the control group (Figure 2E,F). These results indicated that knockdown of the SHCBP1 expression blocked the G1/S transition of the cell cycle and inhibited B16 cell proliferation. Similarly, the results of the scratch assay showed that the proliferation rate of B16 cells was significantly reduced at 36 h and 48 h after SHCBP1 knockdown (Figure 2G,H).

In addition, we overexpressed SHCBP1 in B16 cells, and the expression level of SHCBP1 increased 2.6 times (Figure 3A,B). The proliferation capacity of B16 cells as well as the velocity of clone formation also increased significantly (Figure 3C,D and Appendix A) and the rate of EdU positive cells increased by 20% compared with the control group (Figure 3E,F). Furthermore, after over-expression of SHCBP1 in the wound healing assay, the cell proliferation rate was also significantly accelerated at 36 h and 48 h (Figure 3G,H). These results indicated that high SHCBP1 expression in B16 cells is beneficial for cell proliferation and colony formation.

### 2.3. SHCBP1 Regulated B16 Cell Proliferation-Related Signaling Pathways

To further investigate the molecular mechanisms underlying the promotion of mouse melanoma B16 cell proliferation by SHCBP1, the transcriptome sequencing technique was applied to investigate the possible influences of SHCBP1 on cell signaling pathways and functions. The findings showed that 15,402 genes were expressed in both the control group and the SHCBP1 knockdown group. Compared with the control group, 1150 and 1450 genes were upregulated and down-regulated in the SHCBP1 group, respectively (*p* < 0.05, *n* = 3) (Figure 4A). Finally, 237 differentially expressed genes (DEGs) were identified using FDR < 0.05 and fold change > 2 as the criteria (Figure 4B). Functional annotation was performed for DEGs. GO enrichment showed that the DEGs were mainly enriched in the extracellular matrix (ECM) and extracellular regions. The molecular functions of DEGs included protein binding and components of ECM structures, while the biological processes included cell migration, cell immunity, and cell proliferation (Figure 4C–E). KEGG enrichment analysis demonstrated that the DEGs were mainly involved in cell proliferation and migration-related signaling pathways, including mitogen-activated protein kinase (MAPK) and Rap1 signaling pathways; a cell immunity-related signaling pathway, including a Toll-like receptor, and NOD-like receptor signaling pathways; and apoptosis-related signaling pathway, including cGMP-PKG signaling pathways (Figure 4F). These results indicated that SHCBP1 is extensively involved in the biological processes of B16 cells.

### 2.4. SHCBP1 Accelerates Cell Cycle Progression by Activating the ERK1/2 Signaling Pathway

Preliminary experiments have shown that SHCBP1 promotes the proliferation of B16 cells, but the specific mechanism remains unclear. Therefore, we used knockdown and over-expression of SHCBP1 in B16 cells to analyze the expression of cell proliferation-related proteins. The results showed that p21 expression increased significantly and Cyclin D1 protein expression reduced significantly in cells with downregulated SHCBP1 expression. In contrast, in cells with upregulated SHCBP1 expression, expression of p21 protein was reduced and Cyclin D1 protein was increased (Figure 5A,B). These findings suggestthat SHCBP1 can promote G1/S transition of the cell cycle by inhibiting p21 expression, thus accelerating the process of cell proliferation. Combined with transcriptome sequencing results, these results indicate that SHCBP1 knockdown had a significant effect on the MAPK signaling pathway, in which the expression level of protein tyrosine phosphatase (PTPN7), which inhibits ERK1/2 phosphorylation [20], increased approximately three times in the Ras-MEK1/2-ERK1/2 signaling pathway (Figure 5C). MAPK cascade is a highly conserved pathway that participates in various functions, including cell proliferation, differentiation, and migration. The DEGs in MAPK signaling were further screened (>2-fold change cutoff) for RT-qPCR verification, and the results were in agreement with the sequencing results, which verified the reliability of sequencing findings (Figure 5D). These findings demonstrate that SHCBP1 promoted the proliferation of B16 cells through the MAPK/ERK signaling pathway.

To further confirm the effect of SHCBP1 on the MEK1/2-ERK1/2 signaling pathway, we focused on ERK1/2 phosphorylation. SHCBP1 showed no significant change in total ERK1/2 protein, but the phosphorylation level of ERK1/2 was affected, regardless of whether SHCBP1 was expressed at low or high levels. When SHCBP1 was knocked down, the phosphorylation level of ERK1/2 was significantly reduced, while the phosphorylation level of ERK1/2 was significantly increased after over-expression of SHCBP1, suggesting that SHCBP1 promotes the proliferation of B16 cells through the activation of the ERK1/2 signaling pathway (Figure 5E,F).

### 2.5. Resveratrol Inhibits B16 Cell Proliferation through the SHCBP1/ERK1/2 Signaling Pathway

Previous experiments in this study demonstrated that SHCBP1 promoted the proliferation of B16 cells through the ERK signaling pathway. Given the inhibitory effects of resveratrol on SHCBP1 expression, we hypothesized that resveratrol could regulate the proliferation of B16 cells through SHCBP1. To verify this hypothesis, B16 cells were treated with resveratrol, which significantly reduced the number of cells in the S phase, while the number of cells in the S phase increased significantly in cells over-expressing SHCBP1, which was in agreement with the findings of previous experiments. As previous experiments showed that resveratrol could inhibit the expression of SHCBP1, B16 cells over-expressing SHCBP1 were treated with resveratrol, and the number of S-phase cells increased by 12% compared with the group treated with resveratrol alone, indicating an accelerated cell proliferation rate and suggesting that SHCBP1 may mediate the regulation of B16 cell proliferation by resveratrol (Figure 6A,B).

Compared with SHCBP1 knockdown, the expression of SHCBP1 was further significantly decreased, the expression of the p21 protein was significantly increased, and the expression of Cyclin D1 and ERK1/2 phosphorylation was significantly decreased in resveratrol-treated B16 cells (Figure 6C–G). However, the expression level of Cyclin D1 was significantly increased and p21 protein expression was significantly decreased in resveratrol-treated B16 cells over-expressing SHCBP1, indicating that the cell proliferation rate was accelerated (Figure 7A–D). Furthermore, the phosphorylation level of ERK1/2 was significantly increased (Figure 7E), suggesting that the over-expression of SHCBP1 compensates for the inhibitory effect of resveratrol on cell proliferation, and promotes cyclin expression and cell proliferation by promoting the activation of the ERK1/2 signaling pathway. It was also confirmed that resveratrol inhibited the proliferation of B16 cells by regulating the phosphorylation level of the ERK1/2 signaling pathway by SHCBP1.

## 3. Discussion

As a natural polyphenol, resveratrol showed strong antioxidant and anti-radical biological activities [21]. Notably, resveratrol has anti-proliferative effects [22]. It has been reported that resveratrol can inhibit the migration and invasion of melanoma cells [23]. In this study, resveratrol inhibited G1/S transition in the cell cycle by downregulating the expression of Cyclin D1, resulting in the decreased activity of B16 cells and significantly reducing the cell proliferation rate. Research has shown that resveratrol can effectively reduce the activity of AKT/ERK, PI3K, mTOR and other signaling pathways [24,25,26], promote autophagy in melanoma cells [25], and inhibit the migration and invasion of melanoma cells [27]. In this study, resveratrol significantly reduced the phosphorylation level of the ERK1/2 protein in B16 cells, indicating that resveratrol inhibited the activity of the ERK1/2 signaling pathway. The ERK signaling pathway is a key signal transduction pathway in tumor cells, which is involved in the regulation of various cellular processes, such as cell differentiation, proliferation, and apoptosis [28,29]. Therefore, it is speculated that resveratrol can maintain the stability of the p21 protein in B16 cells by reducing the activity of the MAPK/ERK signaling pathway, inhibiting the expression of Cyclin D1 and blocking the cell cycle in the G1 phase, thus inhibiting cell proliferation. Interestingly, the inhibitory effects of resveratrol on B16 cell proliferation were accompanied by significantly reduced expression of the SHCBP1 protein.

SHCBP1 is a member of the Src homologue and the collagen homologue (SHC) family [30]. In normal human tissues and organs, the mRNA expression level of SHCBP1 is tissue-specific, with the highest expression in testicular tissues [31] and the lowest expression in spleen and lung tissues [15]. SHCBP1 is closely related to embryonic development, growth factor signal transduction, cytokinesis, spermatogenesis, tumorigenesis, and viral infection [32,33]. Previous studies have shown that SHCBP1 is highly expressed in many malignant tumors such as breast cancer [11], lung cancer [34], and glioma [35]. Similarly, our study also found that SHCBP1 was highly expressed in melanoma B16 cells. SHCBP1 promoted the G1/S phase transition in the cell cycle by inhibiting the expression of p21, thus facilitating the proliferation and migration of B16 cells. This means that SHCBP1 is actively involved in the development and progression of melanoma. Peng C [36] demonstrated that SHCBP1 regulated many proteins participating in various signaling pathways, such as the MEK, ERK, and AKT pathways, and effectively activated the MAPK/ERK and PI3K/AKT signaling pathways. Furthermore, transcriptome sequencing analysis showed that SHCBP1 had a greater impact on the MAPK/ERK1/2 signaling pathway. At the cellular level, SHCBP1 significantly activated the ERK1/2 signaling pathway. Notably, activation of the ERK signaling pathway promoted degradation of p21 in normal cells [29]. Taken together, SHCBP1 inhibited p21 transcription by activating the ERK1/2 signaling pathway in B16 cells, thus accelerating the cell cycle transition from the G1 phase to the S phase.

Numerous studies have demonstrated that polyphenols have unique anti-tumor effects by inhibiting cell proliferation, inducing cell apoptosis, regulating the cell cycle, and influencing signal transduction [37,38]. Resveratrol is a polyphenol that downregulates the activity of the NF-κB signaling pathway to reduce the expression of the MMP-9 protein, and thus reduce the invasiveness of HepG2 cells and inflammatory responses in colorectal cancer cells treated with lipopolysaccharide (LPS) [39]. Interestingly, Oroxylin A, a flavonoid, was found to inhibit SHCBP1 expression in synovial sarcoma cells, affecting the activity of the NF-κB signaling pathway and regulating the expression of inflammatory factors that have an inhibitory effect on tumor cell proliferation [40].

In our study, resveratrol was also found to inhibit the expression of SHCBP1, which affects cell proliferation through the ERK1/2 signaling pathway. SHCBP1, as an important upstream protein of the ERK1/2 signaling pathway, may be involved in the process of polyphenols inhibiting tumor cell proliferation. These results suggest that resveratrol inhibits the proliferation of B16 cells by regulating the phosphorylation level of the ERK1/2 signaling pathway through SHCBP1. Our findings were confirmed only in B16 cells and were not validated in other mouse melanoma cell lines, melanoma cell lines from other species, or in vivo animal models.

## 4. Materials and Methods

### 4.1. Cell Lines and Treatment

Mouse melanoma B16 cells were maintained at 37 °C in a 5% CO_2_ humidified incubator and were grown in RPMI-1640 (GIBCO) plus 10% FBS (GIBCO) and 1% penicillin/streptomycin (MP). Mouse melanin precursor cells (MMPCs) were maintained at 37 °C in a 5% CO_2_ humidified incubator and were grown in DMEM (GIBCO) plus 10% FBS (GIBCO) and 1% penicillin/streptomycin (MP).

### 4.2. Cell Viability Assay

The cells to be tested were seeded in 96-well plates at a density of 5 × 10^3^ per well and cultured for 0, 12, 24, 36, and 48 h. Cell viability was assessed by a Cell Counting Kit-8 (Sangon Biotech, Shanghai, China). All data were obtained by the average of five independent experiments.

For the 5-ethynyl-2′-deoxyuridine (EdU) assay, positive cells were observed in red under a fluorescence microscope. (Leica, Wetzlar, Germany).

### 4.3. Clonogenic Assay

Approximately 500 cells were taken from each group and inoculated into 6-well plates at 37 °C for 2 weeks. For crystal violet staining, plates were washed once with PBS and were incubated in 4% paraformaldehyde fixation solution (Boster Biological Technology Co. Ltd., Wuhan, China) for 5 min and then in crystal violet solution for 15 min. Plates were subsequently washed twice with double distilled water, air-dried, and scanned using a Canon scanner.

### 4.4. Cell Migration Assay

When the B16 cells had grown to 100% confluence in a 6-well plate, the monolayer cells were scraped with sterile nozzles to create a small wound. Serum-free medium was added, photos were collected by microscope every 12 h, the width of the scratch was measured, and the gap width of the wound was measured by Image J software (ImageJ 1.51j8) (http://imagej.nih.gov/ij).

### 4.5. Flow Cytometry

A total of 5 × 10^6^ cells were seeded in 6-well plates for cell cycle analysis. After collection, cells were fixed with 75% ethanol overnight at 4 °C and then stained with propidium iodide for 30 min at 37 °C in the dark. ≥10,000 events were analyzed with the use of a BD flow cytometer (FACSAria) and FlowJo software (version 7.6.1) for data analysis.

### 4.6. RNA Isolation and qRT-PCR

The total RNA of the B16 cell line was extracted with Trizol reagent (Invitgen, Carlsad, CA, USA). Reverse transcription of the RNA into cDNA was performed according to the instructions of the Prime Script RT Reagent Kit with a gDNA Eraser. The mRNA expression level was detected by a real-time fluorescence quantitative polymerase chain reaction, and β-actin was used as an internal reference gene. The real-time quantitative PCR procedure was as follows: 94 °C for 10 min, 95 °C for 15 s, 60 °C for 30 s, and 40 cycles.

### 4.7. Western Blotting

Briefly, total proteins were extracted from B16 cells using radioimmunoprecipitation lysis buffer. Then, the samples were separated by 10% sodium dodecyl sulfate–polyacrylamide gel electrophoresis and electrotransferred onto a poly (vinylidene fluoride) membrane. After blocking the membranes in 5% nonfat dry milk, the membranes were immunoblotted with specific primary antibodies (SHCBP1, p21, p27, p53, Cyclin D1, t-ERK, p-ERK, GAPDH) and incubated overnight at 4 °C. The membranes were washed and conjugated with a horseradish peroxidase-conjugated secondary antibody. Immunoreactivity was determined with an ECL Prime Western Blotting Detection System (GE Healthcare).

### 4.8. SHCBP1 Interferes with Cell Line Construction

Design shRNA interference sequence of SHCBP1 (SHCBP1-F5′-CCGGTGATCTGTTGTCTG-GTATAAACTCGAGTTTATACCAGACAACAGATCATTTTTG-3′; SHCBP1-R5′-AATTCAAAAA-TGATCTGTTGTCTGGTATAAACTCGAGTTTATACCAGACAACAGATCA-3′), the interference vector targeting SHCBP1 was constructed, and the mouse B16 cells were infected with lentivirus, and the SHCBP1 stable knockdown cell line was obtained by puro screening. 

### 4.9. Transcriptome Sequencing Data Analysis

Cells of each group were collected for sequencing analysis. The sequencing was completed by Beijing BGI Co., Ltd., (Beijing, China) and the sequencing system was based on an Illumina sequencing platform. In this study, GO and KEGG enrichment analysis was performed using an OmicShare online data analysis platform tool (www.omicshare.com/tools accessed on 5 November 2023).

## Figures and Tables

**Figure 1 molecules-28-07614-f001:**
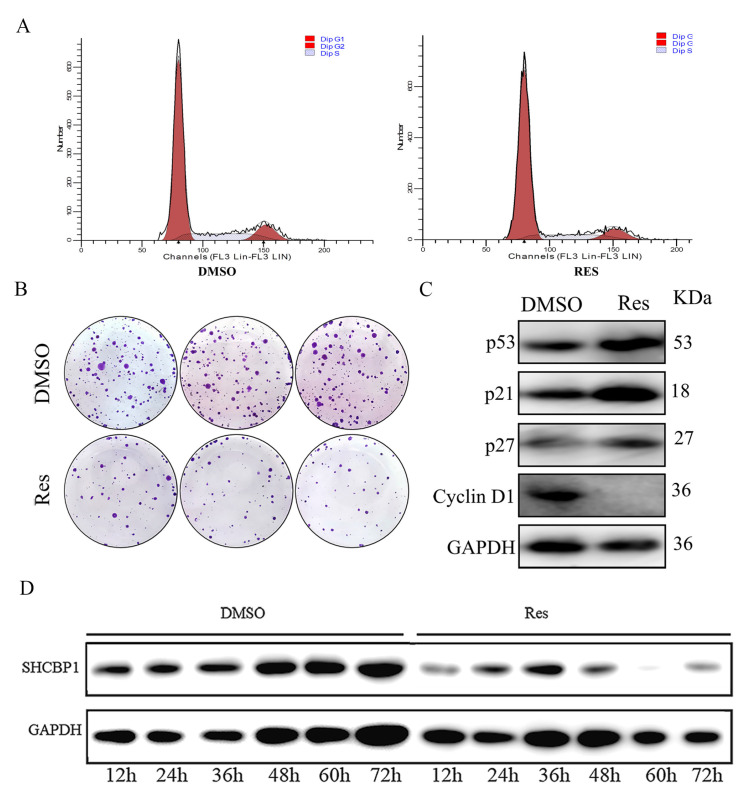
Resveratrol inhibits B16 cell proliferation and SHCBP1 protein expression. B16 cells were treated with different concentrations of resveratrol for 48 h. (**A**) The number of G1, S and G2 cells in B16 cells was detected by flow cytometry, and DMSO was added as the control group. (**B**) Crystal violet fuel would be used to stain the subclonal cells formed after resveratrol treatment for quantitative statistics of cell cloning. (**C**) The expression of p21, p27, p53 and Cycling D1 proteins in B16 cells was detected by Western blotting for 48 h after resveratrol addition, *n* = 3. (**D**) Protein imprinting was collected every 12 h to detect the expression of SHCBP1 in B16 cells, *n* = 3.

**Figure 2 molecules-28-07614-f002:**
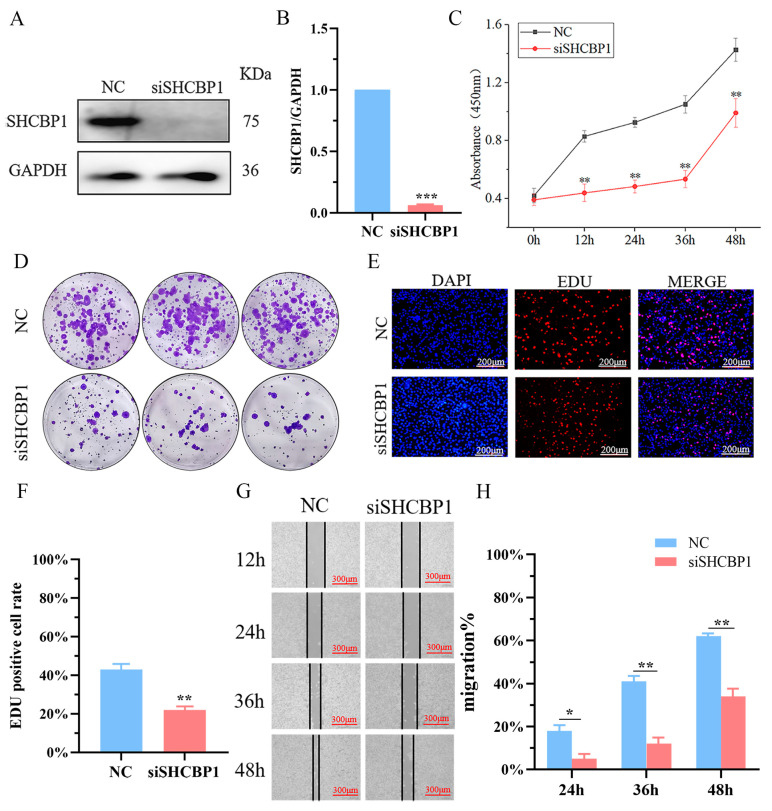
NC: control group; siSHCBP1: Lower the SHCBP1 group. (**A**,**B**) The protein expression level of SHCBP1 with the interference efficiency was detected by Western blotting. *** *p* < 0.001, *n* = 3. (**C**) Cell proliferation of B16 was measured by CCK8 assay at 0, 24, 48, 72 h post-transfection, respectively. ** *p* < 0.01. (**D**) The clonal formation ability and quantification results of B16 cells were measured after continuous culture for 13 days. (**E**,**F**) The proliferation rate of B16 cells was detected by EDU method. Red: proliferating cells, blue: nucleus, ** *p* < 0.01. (**G**,**H**) Cell wound scratch assay was used to detect cell proliferation after SHCBP1 interference * *p* < 0.05; ** *p* < 0.01.

**Figure 3 molecules-28-07614-f003:**
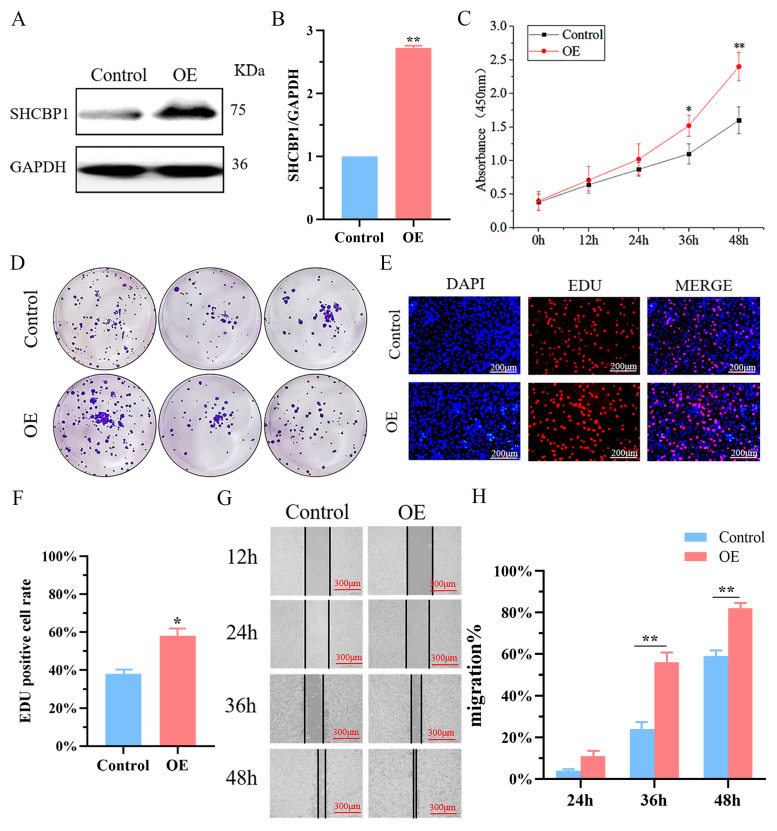
Overexpression of SHCBP1 can promote the proliferation and migration of B16 cells. Control: no-load group; OE: Overexpression group. (**A**,**B**) The protein expression level of SHCBP1 was detected by Western blotting. ** *p* < 0.01, *n* = 3. (**C**) Cell proliferation of B16 was measured by CCK8 assay at 0, 24, 48, 72 h post-transfection, respectively. * *p* < 0.05; ** *p* < 0.01. (**D**) The clonal formation ability and quantification results of B16 cells were measured after continuous culture for 7 days. (**E**,**F**) The proliferation rate of B16 cells was detected by EDU method. Red: proliferating cells, blue: nucleus. * *p* < 0.05. (**G**,**H**) Cell wound scratch assay was used to detect cell proliferation after overexpression of SHCBP1. ** *p* < 0.01.

**Figure 4 molecules-28-07614-f004:**
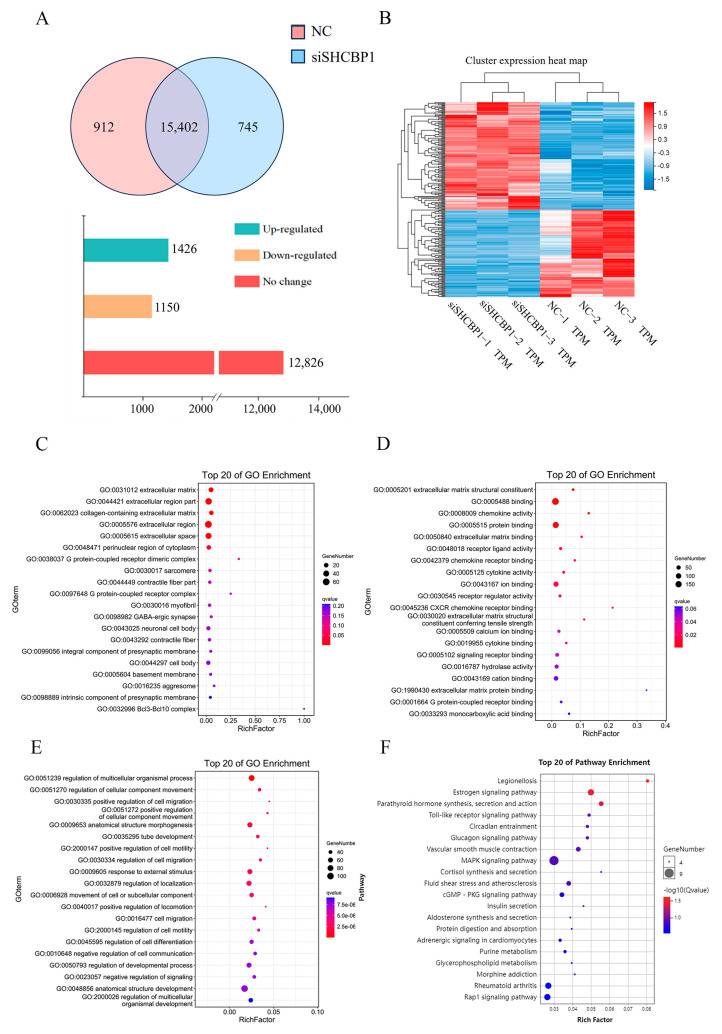
SHCBP1 regulates B16 cell proliferation through MAPK signaling pathway. NC: control group; siSHCBP1: Lower the SHCBP1 group. (**A**) Differential gene expression analysis of control group and SHCBP1 knockdown group. (**B**) Differential gene screening (FDR < 0.05 and fold change > 2). (**C**–**E**) GO enrichment analysis of differential genes. (**F**) KEGG enrichment analysis of differential genes.

**Figure 5 molecules-28-07614-f005:**
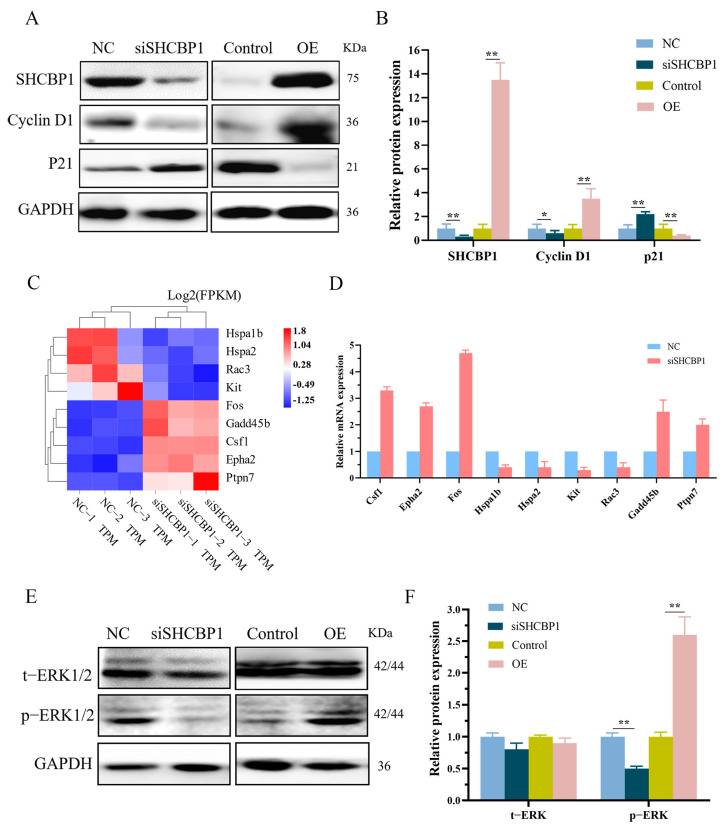
SHCBP1 promotes B16 cell proliferation through ERK phosphorylation, NC: control group; siSHCBP1: Lower the SHCBP1 group. Control: no-load group; OE: Overexpression group. (**A**,**B**) Expression of cell proliferation-related proteins was detected. * *p* < 0.05; ** *p* < 0.01, *n* = 3. (**C**) Heat map analysis of differential genes (>Log2 = 1) in MAPK signaling pathway. (**D**) The expression levels of MAPK signaling pathway-related genes were detected by RT-qPCR. (**E**,**F**) Expression of ERK pathway-related proteins was detected. ** *p* < 0.01, *n* = 3.

**Figure 6 molecules-28-07614-f006:**
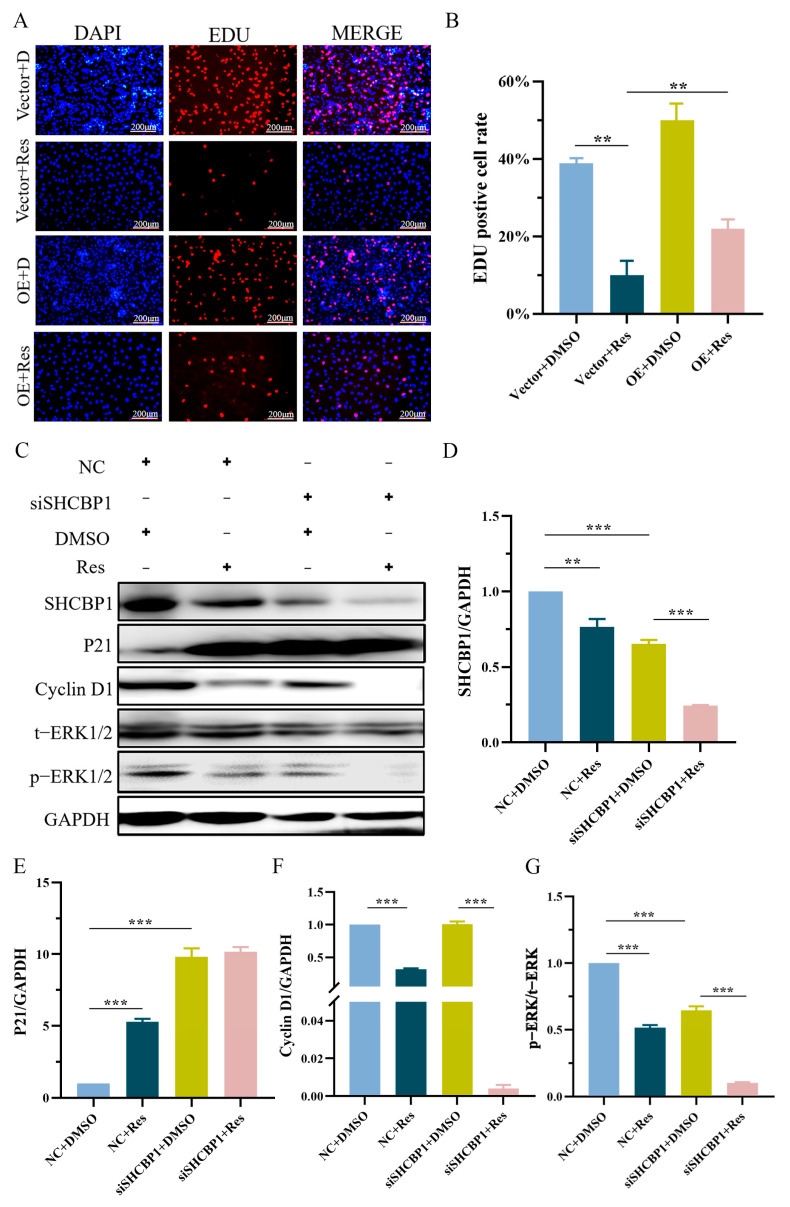
SHCBP1 knockdown enhanced the inhibitory effect of resveratrol on B16 cell proliferation through ERK signaling pathway. Four experimental groups were set up. No load + dimethyl sulfoxide (Vector + DMSO), no load + resveratrol (Vector + Res), knockdown + dimethyl sulfoxide (siSHCBP1 + DMSO), knockdown + resveratrol (siSHCBP1 + Res). (**A**,**B**) B16 cells were treated with resveratrol and cell proliferation rate was measured by EDU. ** *p* < 0.01. (**C**–**G**) After SHCBP1 knockdown, B16 cells were treated with resveratrol for 48 h and Western blotting was used to detect the expression of SHCBP1, Cycling1, p21, p-ERK and other proteins. ** *p* < 0.01; *** *p* < 0.001, *n* = 3.

**Figure 7 molecules-28-07614-f007:**
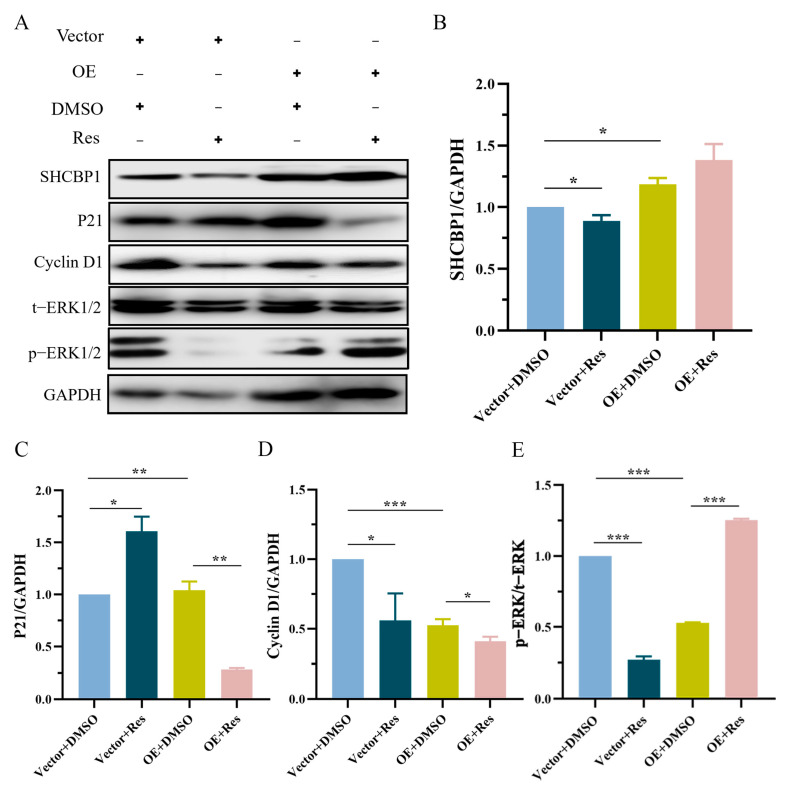
SHCBP1 compensates for the inhibitory effect of resveratrol on B16 cell proliferation through ERK signaling pathway. Four experimental groups were set up. No load + dimethyl sulfoxide (Vector + DMSO), no load + resveratrol (Vector + Res), overexpression + dimethyl sulfoxide (OE + DMSO), overexpression + resveratrol (OE + Res). (**A**–**E**) After overexpression of SHCBP1, B16 cells were treated with resveratrol for 48h and Western blotting was used to detect the expression of SHCBP1, Cycling D1, p21, p-ERK. * *p* < 0.05; ** *p* < 0.01; *** *p* < 0.001, *n* = 3.

## Data Availability

Data are contained within the article.

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
