# Peer review of "Effects of Resveratrol on Mouse B16 Melanoma Cell Proliferation through the SHCBP1-ERK1/2 Signaling Pathway"

_molecules, 2023, doi:10.3390/molecules28227614_

Round 1

Reviewer 1 Report

Comments and Suggestions for Authors

Major Comments:

1.     The studies involve in vitro analyses only, hence, would analogous findings be discerned in progressively growing melanomas? The authors must validate their study in in vivo animal models.

2.     The studies involve the analysis of a single murine melanoma cell line which limits conclusions related to generality of findings.

3.     When analyzing the quantified data sets, you have to include all data (technical triplicates AND repeated experiments)

4.     All graphs in figures should represent the individual values.

5.     Were the cell lines tested for mycoplasma?

Minor Comments:

1.     Please check the manuscript for grammatical errors and typos.

Comments on the Quality of English Language

Please check the manuscript for grammatical errors and typos.

Reviewer 2 Report

Comments and Suggestions for Authors

This manuscript investigates the mechanism of resveratrol in the regulation of melanoma progression and finds that the SHCBP1/MAPK-Erk axis is crucial for the anti-melanoma activities of resveratrol. There are a couple of flaws that the authors need to address.

Major Critiques:

1.       Line 59: Isn’t SH2 the phospho-tyrosine binding domain? The reference 14 says phospho-serine based on two citations. However, both of the citations described SH2 as phospho-tyrosine binding domain and it was miswritten in the reference 14 and again by the authors in this manuscript.

2.       Line 58-61: The whole sentence is exactly the same (copied and pasted) as it is in the reference. The author should rephrase with your own words.

3.       In Fig 1A, which one is treated by resveratrol? Control group is usually plotted on the left side. However, it looks like the left one has G1 arrest and the right figure has more cells in G2/S phases for cell cycle progression. Please clarify this confusion.

4.       In the supplementary figure legend 1, what does the protein imprinting refer to and how is protein imprinting used for protein expression and quantification purposes?

5.       In the supplementary Figure 2D, please define “OE”. Additionally, since the sample number is only 3 (n=3), it is obviously not statistically significant given the overlapped error bar. Please check your statistical analyses.

6.       The cell scraping/wound healing assay in Fig 2G and H are typically not definitive experiments for cell migration measurement. As SHCBP1 knockdown blocks B16 proliferation, the narrowing of scraped area could be just a reflection of slower growth to fill up the gap. Another solid evidence should be provided if the authors wish to claim the impact on cell migration by SHCBP1 knockdown. The same is true for Figure 3G and H.

7.       What is the siRNA knockdown efficiency? In figure 2, it was 94% knockdown efficiency. However, it was ~30% in Figure 6D (yellow bar). What causes this inconsistency ?

8.       How long was the cells treated with resveratrol in Figure 6? The SHCBP1 expression by resveratrol treatment varies (Fig 1D). More details would help the interpretation of the results.

9.       The whole work was done using only one cell line, B16. This is a serious concern whether the findings can be reproduced in different melanoma cell lines. If the authors do not use other cell lines, it should be clarified that these findings are only seen in B16 cells, and unknown for any other melanoma cells.

Minor critiques:

1.       Line 137, do you mean Figure 3 C and D?

2.       Line 327: a fluorescence. Please check the space between words.

3.       Line 358: polyacrylamide gel electrophoresis. Please check the space in and between words.

Comments on the Quality of English Language

see comments in the section above.

Round 2

Reviewer 1 Report

Comments and Suggestions for Authors

Authors have not satisfactory addressed the previous concerns. 

Comments on the Quality of English Language

Moderate editing of English language required

Reviewer 2 Report

Comments and Suggestions for Authors

The authors appropriately addressed most concerns. As for the single cell line usage, I would recommend to rephrase the title as "Effects of Resveratrol on mouse B16 melanoma cell proliferation through the SHCBP1-ERK1/2 signaling pathway" if no other supported results from other cell lines/animal models are presented. 

Alternatively, the authors may upload the results using other cell lines as supplementary figures even though less significant effect was noted. 

Comments on the Quality of English Language

readable and understandable but there's room for improvement.

Round 3

Reviewer 1 Report

Comments and Suggestions for Authors

Authors have addressed the previous concerns. 

Comments on the Quality of English Language

Minor editing of English language required